# Research on Adhesive Coefficient of Rubber Wheel Crawler on Wet Tilted Photovoltaic Panel

Minh Tri Nguyen [1], Cong Toai Truong [1,2,3], Vu Thinh Nguyen [1,2,3], Van Tu Duong [1,2,3], Huy Hung Nguyen [1,4] and Tan Tien Nguyen [1,2,3,*]

1. National Key Laboratory of Digital Control and System Engineering (DCSELab), Ho Chi Minh City University of Technology (HCMUT), 268 Ly Thuong Kiet Street, District 10, Ho Chi Minh City 700000, Vietnam; tri.nguyen1710349@hcmut.edu.vn (M.T.N.); 2298001@hcmut.edu.vn (C.T.T.); vuthinh.n@hcmut.edu.vn (V.T.N.); dvtu@hcmut.edu.vn (V.T.D.); nhhung@dcselab.edu.vn (H.H.N.)
2. Faculty of Mechanical Engineering, Ho Chi Minh City University of Technology (HCMUT), 268 Ly Thuong Kiet, District 10, Ho Chi Minh City 700000, Vietnam
3. Vietnam National University Ho Chi Minh City, Linh Trung Ward, Thu Duc District, Ho Chi Minh City 700000, Vietnam
4. Faculty of Electronics and Telecommunication, Saigon University, Ho Chi Minh City 700000, Vietnam
* Correspondence: nttien@hcmut.edu.vn; Tel.: +84-918-255-355

**Abstract:** The demand for renewable energy sources is growing fast because of the negative impact of the utilization of fossil energy, nuclear energy, and hydroelectricity. One of the renewable energy sources, known as solar energy, which uses the photovoltaic panel (PV) to generate electricity from the sun, is a promising alternative that has great potential to deal with the power crisis. However, the power productivity and efficiency conversion are affected significantly by dust accumulation on PVs. Many researchers investigated PV panel dust cleaning methods to improve performance, yield, and profitability. Various dust cleaning and mitigation methods such as rainfall, labor-based, and mechanized cleaning are explored, and we demonstrated that dust removal could be automated with cleaning robots effectively. Due to the specified geographical site of PV panel installation, cleaning robots might work on the misalignment and uneven PV arrays, presenting huge challenges for an autonomous cleaning robot. Thus, a rubber wheel crawler robot with semi-autonomous handling provides a flexible motion that is a well-suited solution to clean rooftop PV arrays. Nevertheless, the rubber wheel crawler robot might suffer slippage on the wet glass of tilted PV arrays. This paper studies the anti-slip effect of the rubber wheel crawler equipped with a cleaning robot under the wet surface of tilted PV panels. First, a theoretical model consisting of several parameters is established to validate the slippage of the rubber wheel crawler on the wet tilted PV. Then, some parameters of the theoretical model are approximated through experimental tests. Finally, simulation results of the theoretical model are conducted to evaluate the accuracy of the proposed theoretical model in comparison to the experimental results under the same working conditions. The merits provide the efficient design of rubber wheel crawlers, enabling the anti-slip ability of robots.

**Keywords:** solar panel cleaning robots; wet surface; shore hardness; synthetic rubber; photovoltaic panel

## 1. Introduction

The vogue of smart home technology [1] alongside intelligent compliances and the development of industry 4.0 [2] gaining production capabilities [3] due to computerization manufacturing, cloud computing, and robotic systems [4,5] have increased the demand of energy resources. However, the fossil energy sources are quickly exhausted [6] and cause environmental pollution [7]. Meanwhile, nuclear energy might harm not only humans but also animals and plants, as it causes mutations in DNA [8,9], and the habitat takes a long time to recover [10]. This energy crisis has led to the introduction of renewable

energy as a potential alternative [11]. Recently, solar energy, which generates electricity using the photovoltaic (PV) panel, has increased exponentially in many developed and developing tropical countries around the world [12,13]. The growth of solar energy is indicated by the increasing installed capacities in both rooftop PV panel arrays for small-scale applications and concentrated PV panel arrays for mainstream power [14–16]. Among all the other renewable energy sources, solar energy systems have many advantages in terms of flexible installation sites, cheap maintenance and installation costs, and being environmentally friendly. However, the efficient conversion of the solar energy system is decreased significantly by two merits parameters: cloudy days and accumulated dust [17]. Additionally, the longevity of PV panels is affected by heating induced on dirty PV cells.

Some researchers have studied the degradation of PV panels output [18–20] under the accumulated dust. In Europe and the Middle East, the efficiency of dirty PV panels decreases by 3–6% per year, while it has decreased by more than 30% in India and the Arabian Peninsula [21–23]. For tropical countries such as Thailand and Vietnam, this is problematic, with the efficiency of solar cells decreased by 35% [24]. The work [25] has pointed out that a clean PV panel can improve performance considerably and yield more than the dirty one. Some of the cleaning methods for PV panels, such as rainfall cleaning [26–28], labor-based cleaning [29], and mechanized cleaning [30,31], were introduced in the past. The rainfall cleaning method requires a large tilted angle of PV panels that are not costly to clean and mitigate loose dust on the PV surface. However, the disadvantage of this method is that it depends on the rainfall intensity at the installation site. If the rainfall is insufficient, it will cause sludge formation on the PV panel, or only large particles will be removed [32]. Thus, the rainfall cleaning method has low efficiency and is mainly adopted in areas where it is hard to reach for maintenance [33,34]. The labor-based cleaning method requires costly time, human resources, water, and cleansing facilities [35]. This cleaning method can improve the conversion efficiency and cool down the PV panel [36–40], but it is an expensive cleaning cost compared to the rainfall cleaning method [41]. Furthermore, cleansing facilities might scratch the dry PV panel surfaces [35]. As well, human workers cannot maintain their focus on work due to declining physical conditions under the prolonged cleaning time. Mechanized cleaning methods employ motorized cleaning brushes or portable cleaning robots [42–46] and water supply to clean the PV panel surface [47]. The study [48] pointed out that this cleaning method should be applied to clean the PV panels once per week or daily when the rate of dust is intense. The benefit of the mechanized cleaning method is to cool PV panels with water, use automated operation by cleaning robots, and save human resources [49]. Alternatively, the PV cleaning robot can be classified into two categories. The first is to use two motorized trolley carriers clamping on the top and bottom edge of the PV panel [15] to move throughout the PV panel arrays, which is well-suited for mainstream power with massive concentrated PV panel arrays. The other is based on the structure of rubber wheel crawler robots, which is preferred for small-scale PV systems [29]. The advantages of this type of robot are portability and the ability to work with misalignment and uneven surfaces of PV panels. The drawback is that it requires human teleoperation, and the cleaning robot might slip on a wet tilted PV panel.

In Vietnam, the rooftop PV system has grown exponentially, with the capacity installations adding 9.3 gigawatts (GW) to its national grid at the end of 2020 compared to that of 378 megawatts (MW) in 2019. This leads to the increasing demand for cleaning accumulated dust on the rooftop PV panels. As aforementioned, a rubber wheel crawler robot with the anti-slip ability on the wet large tilted angle of PV panels seems to be the well-suited solution. This paper studies the anti-slip effect of a rubber wheel crawler equipped with cleaning robots on the wet large tilted angle of the rooftop PV panel. The objective of this study is to establish the relationship between the friction coefficient of the rubber wheel crawler and the surface of the PV panel with respect to various conditions such as the tilted angle of the PV panel, rubber hardness, and dry/wet surfaces.

## 2. Materials and Methods

The cleaning robot which is used to clean the accumulated dust on the glass surface of the PV panel arrays consists of three modules: two rotating brushes with a water supply and a chassis module equipped with two rubber wheel crawlers that contains an electrical enclosure and battery. The rubber wheel crawler is driven by two primary timing pulleys through a timing belt coated with rubber trackpads. The PU timing belt is supported by auxiliary free wheels that can hump when moving on the misalignment/uneven surface, thanks to the spring tensioners. The rubber wheel crawlers provide the cleaning robot the ability to move on the PV panel surfaces longitudinal to its chassis as well as rotate in place. The usage of the rubber wheel crawler enables a larger area of contact between the cleaning robot and the PV panel surface than the usage of traditional wheels, leading to increased friction force. For the slightly tilted angle of the PV panel (less than 10°), there is no/rarely slip on the rubber wheel crawler and the PV panel surface; the cleaning robot moves along/orthogonal to the slope of PV panels. However, when there are PV panels with a large tilted angle (over 10°) in wet conditions, this is a major challenge for researchers to increase the slip resistance between the rubber track and PV panel surface. According to [27,50–54], the tilted angle of the PV panel varies $0° \div 60°$ due to the geographical installation site. The larger tilted angle of PV panels leads to the lower slip resistance at the rubber wheel crawler and PV panels. There are many factors that affect the slip resistance between the rubber wheel crawler and PV panel surfaces: for instance, spring forces, the area of contact, and the weight of the cleaning robot. However, this paper mainly investigates the characteristics of the rubber materials coated on the trackpad of the timing belt to improve the slip resistance by optimizing the reaction force.

Assumption:

- There is no slip on the primary timing pulleys and the PU timing belt;
- The tensions of the PU timing belts on both sides of the cleaning robot are the same;
- The slip resistance is not affected by the direction of rotation of the broom

Consider the cleaning robot in the stationary state on the PV panel with a tilted angle $\alpha$ ($0° \leq \alpha \leq 60°$), as shown in Figure 1. The cleaning robot is applied by the cleaning robot's gravity ($\overrightarrow{P}$), the reaction force ($\overrightarrow{N}$), the static friction force between the belt and the panel surface ($\overrightarrow{F_{fr}}$) [55–60], and the elastic force of the timing belts ($\overrightarrow{F_E}$). According to the no-slip condition, the greater the static friction, the larger the slip resistance. On the other hand, the static friction force must be larger than the cleaning robot's gravity to ensure the no-slip condition of the cleaning robot.

$$\overrightarrow{F_{fr}} \geq \overrightarrow{P} \tag{1}$$

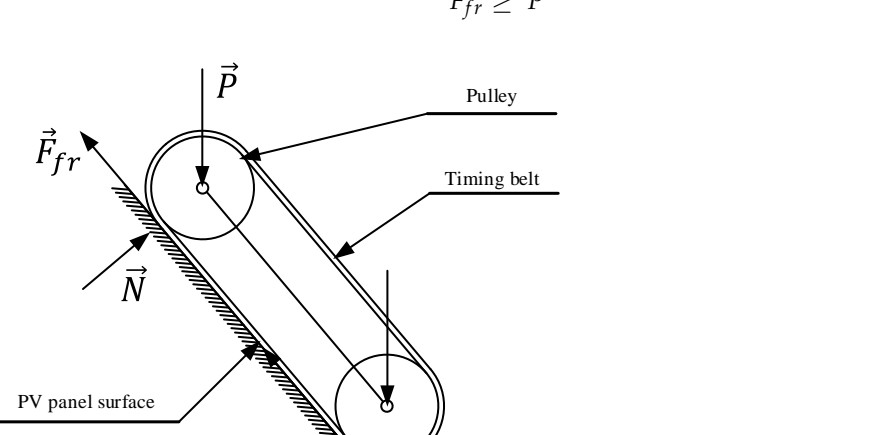

**Figure 1.** Force analysis in the no-slip state and zero acceleration.

With the tilted angle $\alpha$ of the PV panel, the amplitude of the cleaning robot's gravity projected into the shopfloor is calculated by the term of $P \sin \alpha$. The amplitude of the static friction force is a product of the coefficient of friction ($\mu$) and reaction force ($N$) [61,62]. Equation (1) described the adhesive condition between the rubber track and PV panel surface.

$$\mu N - P \sin \alpha \geq 0 \tag{2}$$

It can be seen from Equation (2) that the adhesive condition depends on the reaction force ($N$). The larger the reaction force, the higher the slip resistance. It is evident that the reaction force increases as the cleaning robot's weight increases; however, this is impossible because the weight of the cleaning robot should be minimized to avoid the overload of the PV panel.

Figure 2 describes the equivalent model of the rubber trackpad coating belt. The elastic characteristic of the rubber trackpad is denoted by springs in parallel with the spring stiffness ($K$) and the displacement ($\Delta L$) that produce the elastic force ($\overrightarrow{F_E}$) in the opposite of the direction of the cleaning robot's gravity. Applying Newton's third law, the relationship between the reaction force, the cleaning robot's gravity, and the elastic force can be expressed as follows:

$$\overrightarrow{N} = \overrightarrow{P} - \overrightarrow{F_E} \tag{3}$$

where the elastic force ($F_E$) is a product of the spring stiffness ($K$) and the displacement ($\Delta L$) according to Hooke's law that can be calculated as

$$F_E = K \Delta L \tag{4}$$

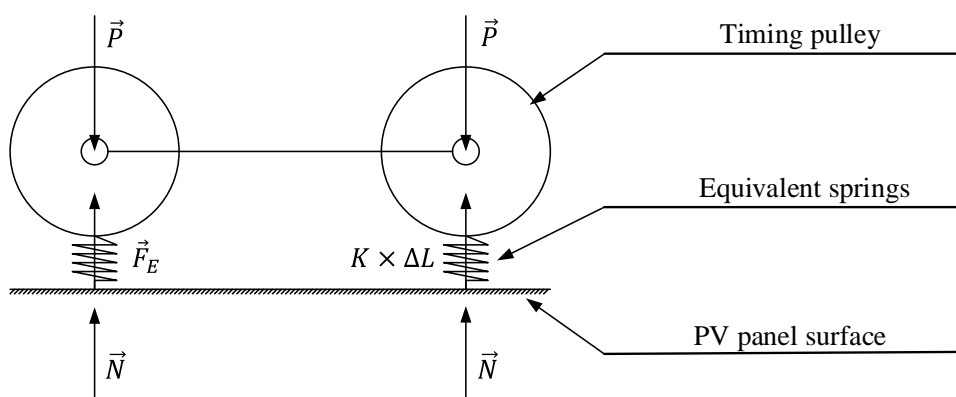

**Figure 2.** Equivalent model of rubber track pad coating belt.

By using the projection of Equation (3) into the perpendicularity of the ground, Equations (2) and (4) can be rewritten as follows:

$$((P \cos \alpha - K \Delta L)\mu - P \sin \alpha) \geq 0 \tag{5}$$

Equation (5) implies that the spring stiffness ($K$) and the displacement ($\Delta L$) of the equivalent model and the static friction coefficient ($\mu$) should be known to determine the adhesive bonding of the cleaning robot and PV surfaces. These parameters strongly depend on the hardness of the rubber material coating on the rubber track belt. Similar to the material used for the tires, it is made of rubber with variable hardness, according to [63]. From the perspective of anti-slip, the key feature of each type of material is the coefficient of friction between its contact surface and the PV panel surface. According to the experimental research [64], the static friction coefficient ($\mu$) is a function of shore hardness. The function $\mu$ is simplified in the linear form as follows:

$$\mu \approx -xS + y \tag{6}$$

where $x$ is the slope of the function $u$; and $y$ is the maximum coefficient of friction that is determined by practical experiments through the utilization of zero shore harness material.

The spring stiffness ($K$) and the displacement ($\Delta L$) of the equivalent model of a rubber track belt can be determined based on the principle of a measurement device called a durometer, which is used to measure the hardness of materials. The durometer consists of an indentor with the standard weight and a calibrated spring, as shown in Figure 3a. The equivalent model of the durometer can be described in Figure 3b. The material with a thickness ($L_0$) and shore hardness ($S$) can be represented by a spring with the stiffness ($K_{SB}$) and the displacement ($\Delta L_{SB}$).

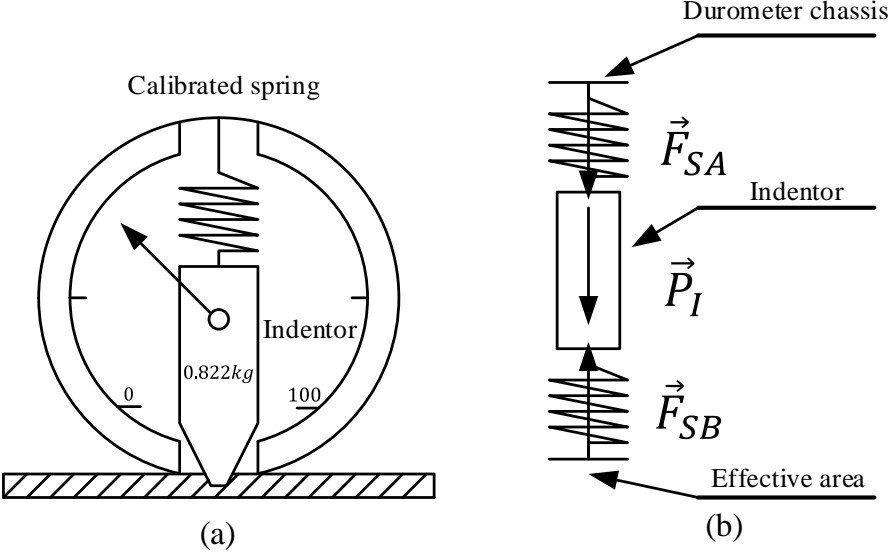

(a)  (b)

**Figure 3.** Basic scheme of a shore durometer (**a**) and the equivalent model of its principle (**b**).

Applying a Free Body Diagram to the equivalent model of a durometer, the sum of gravity of the indentor ($P_I$), the shore above force of the calibrated spring ($F_{SA}$) and the shore below force of the equivalent spring ($F_{SB}$) can be calculated by:

$$\overrightarrow{F_{SA}} + \overrightarrow{P_I} + \overrightarrow{F_{SB}} = 0 \tag{7}$$

It can be seen from Figure 3b that the shore below force of the equivalent spring ($F_{SB}$) is opposite to the other two forces, thus:

$$F_{SA} + P_I = F_{SB} \tag{8}$$

The gravity of the indentor with the standard mass ($m$) of 0.822 kg is calculated as $P_I = mg$, where $g$ is the gravity acceleration of 9.81 m/s$^2$.

The shore above force ($F_{SA}$) can be simplified as a linear function depending on the shore hardness ($S$) [63]:

$$F_{SA} = 0.550 + 0.075S \tag{9}$$

The shore below force of the equivalent spring is a product of the spring stiffness ($K_{SB}$) and the displacement ($\Delta L_{SB}$):

$$F_{SB} = K_{SB}\Delta L_{SB} \tag{10}$$

According to [63], the displacement ($\Delta L_{SB}$) is inversely proportional to the shore hardness expressed in millimeters:

$$\Delta L_{SB} = 0.0254\frac{100 - S}{100} \tag{11}$$

Substituting Equation (10) into Equation (8) yields:

$$K_{SB} = \frac{F_{SA} + P_I}{\Delta L_{SB}} \tag{12}$$

Equations (7)–(12) are only used to determine the equivalent spring stiffness in the case of the small cross-section of the indentor top interacting with the effective area ($a$). For the case of the rubber trackpad in contact with the PV surface, the effective area ($A$) is much more than ($a$). The effective area ($A$) is the product of maximum length ($l_{max}$) and width ($w_{max}$) of the rubber wheel crawler (Figure 4).

$$A = l_{max} w_{max} \tag{13}$$

The equivalent spring stiffness of the rubber track belt is calculated by integrating the small stiffness values ($K_{SB}$) for each effective area ($a$) as below:

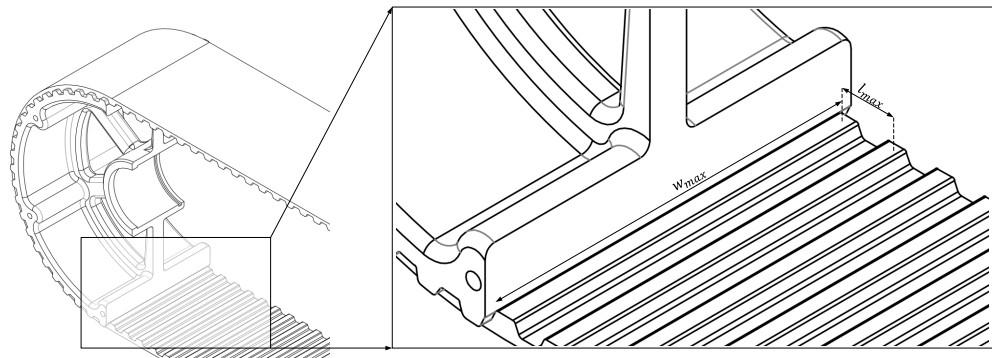

**Figure 4.** CAD model of rubber tracking belt and timing pulleys.

$$K = \int_0^{l_{max}} \int_0^{w_{max}} K_{SB} \frac{A}{wl} dw dl \tag{14}$$

It can be seen from Equation (14) that the equivalent spring stiffness ($K$) depends on the efffective area ($A$) and the maximum length ($l_{max}$), which are strongly affected by the shore hardness.

Similarly to the characteristic of the equivalent spring stiffness ($K$), the displacement ($\Delta L$) of the rubber track belt depends on the shore hardness, the effective area, the applied force, the initial thickness of the material layer, and Young's stress module ($E$).

According to the references [65,66], the correlation between Young's stress module ($E$) and the shore hardness is expressed by:

$$E = \frac{0.0981(56 + 7.62336S)}{0.137505(254 - 2.54S)} \tag{15}$$

On the other hand, Young's stress module can be derived by Hooke's law [67]:

$$E = \frac{FL_0}{A\Delta L} \tag{16}$$

where $F$ is the gravity force reacting on the effective area $A$, and $L_0$ is the initial thickness of the specimen.

Substituting Equation (15) into Equation (16) yields

$$E = \frac{FL_0}{A} \frac{0.137505(254 - 2.54S)}{0.0981(56 + 7.62336S)} \tag{17}$$

Equation (17) shows that there is a correlation between the effective area ($A$) and the displacement ($\Delta L$). The effective area $A$ is formed by the gravity force of the cleaning robot reacting to the contact area between the timing pulley and the rubber track belt (Figure 5).

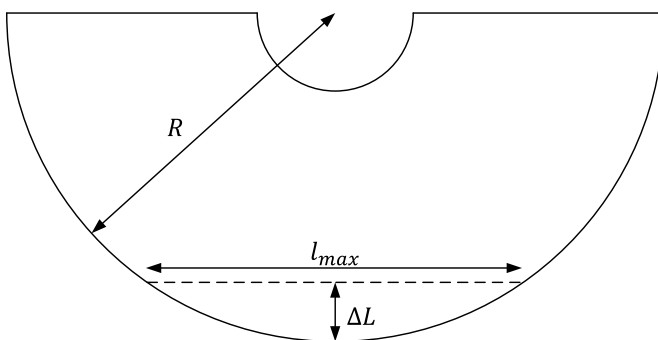

**Figure 5.** The displacement ($\Delta L$) of rubber track belt.

$$A = l_{max}w_{max} = 2w_{max}\sqrt{R^2 - (R - \Delta L)^2} \tag{18}$$

where $R$ is the radius of the timing pulley.

By substituting Equation (18) into Equation (16), it can be deduced that

$$\Delta L = \frac{FL_0}{2Ew_{max}\sqrt{2R\Delta L - \Delta L^2}} \tag{19}$$

## 3. Experimental Study

The logic of this study has the sequence of first including the introduction of the theoretical model. Then, some parameters of this model are obtained through experiment and computational theory. Finally, the relationship between the shore hardness, adhesive coefficient, and the maximum tilted angle of PV is established by simulation studies.

A prototype of a rubber wheel crawler is made and equipped with a PV cleaning robot, as shown in Figure 6. The rubber wheel crawler comprises a seamless polyurethane (PU) timing belt with a T10 tooth profile, which runs over two-timing pulleys. The PU timing belt is 75 mm wide, of length 1690 mm, and it has a rubber coating with a cuboidal back-profile to gain the adhesive coefficient of the robot on the slippage surface of PV. The physical parameter of the rubber wheel crawler is designed as shown in Table 1. The PV cleaning robot, which is 80 kg in weight, is designed to move on the tilted angle PV of $0° \div 45°$.

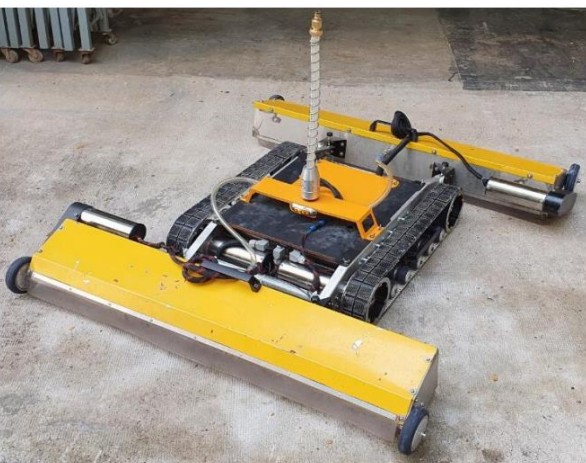

**Figure 6.** Prototype of PV cleaning robot.

**Table 1.** Physical parameters of rubber wheel crawler.

| Parameters | Value | Parameters | Value |
|---|---|---|---|
| Material | Polyurethane (PU) | Coated material | Synthetic/Red rubber |
| Tooth profile | T10 | Coated trackpad | Cuboidal back profile |
| Width | 75 mm | Shore hardness | A35/A40 |
| Length | 1690 mm | Deep recessed | Only block/Zigzag pattern |
| Thinkness | 20 mm | | |

To compare the adhesive coefficient of the coating material, there are two types of rubber materials that are utilized to coat the PU timing belt, which are known as red rubber and synthetic rubber. Both types of rubber wheel crawlers are equipped with the PV cleaning robot for working on the same testing conditions. The shore hardness of both types of rubber is verified by a durometer. The tilted angle of the PV panel can vary $0° \div 45°$, which is measured by an inclinometer, while it is moistened by the stable water flow supply. The PV cleaning robot is required to move on the PV panel transversely with a travel length of 5 m for every $5°$ increasing the tilted angle PV. The slippage state of the PV cleaning robot on the PV surface is defined as the robot being shifted longitudinally to the PV panel by at least 10 cm, which is measurable by using a laser alignment tool.

By conducting experiments on the synthetic rubber track belt with the shore hardness of A35 and the maximum tilted angle of $25°$, the coefficient of friction for the case of using a synthetic coating is determined through Equation (5) as follows:

$$\mu_{syn} \approx \frac{P \sin \alpha}{P \cos \alpha - K\Delta L} = 0.5275 \tag{20}$$

By recalling Equation (6) and assimilating with Equation (20), the range of $x$, $y$ of Equation (6) is determined by referring to the reference [64]:

$$\begin{cases} 1.1431 < y < 2.0494 \\ -0.0435 < x < -0.0176 \end{cases} \tag{21}$$

The values of $x$, $y$ are approximated as follows:

$$\mu_{syn} \approx -0.0305S + 1.5963 \tag{22}$$

In the same manner, the coefficient of friction for the case of using a red rubber coating with the shore hardness of A40 and the maximum tilted angle of $15°$ is determined by

$$\mu_{syn} \approx -0.0315S + 1.5578 \tag{23}$$

For each tilted angle increment and different materials, there exists a certain shore hardness in such a way that can seek the optimal adhesion. Similarly, for each certain shore hardness, there exists a maximum tilted angle value that the cleaning robot can move without slippage.

By using Equations (5), (14), (19) and (22) and referring to Table 1, it can be easy to verify the robot slippage through the below equation:

$$\left( P \cos \alpha - \left( \int_0^{l_{max}} \int_0^{w_{max}} \frac{K_{LB}A}{lw} dw dl \right) \Delta L \right)(-xS + y) - P \sin \alpha \geq 0 \tag{24}$$

The cleaning robot can move on the PV surfaces without slippage if Equation (24) is satisfied; otherwise, it might be slipped.

For a given shore hardness, there exists a maximum tilted angle in such a way that Equation (24) is satisfied. Once the shore hardness is increased, the tilted angle of the PV that the cleaning robot can move without slippage decreases. Figure 7 illustrates the

adhesive coefficient with respect to the tilted angle of the PV that enables the cleaning robot to move without slippage, corresponding to synthetic/red rubber of varying shore hardness in a range of A30–A50. The maximum tilted angle that the cleaning robot can move without slippage with respect to the coating materials is briefly described in Table 2.

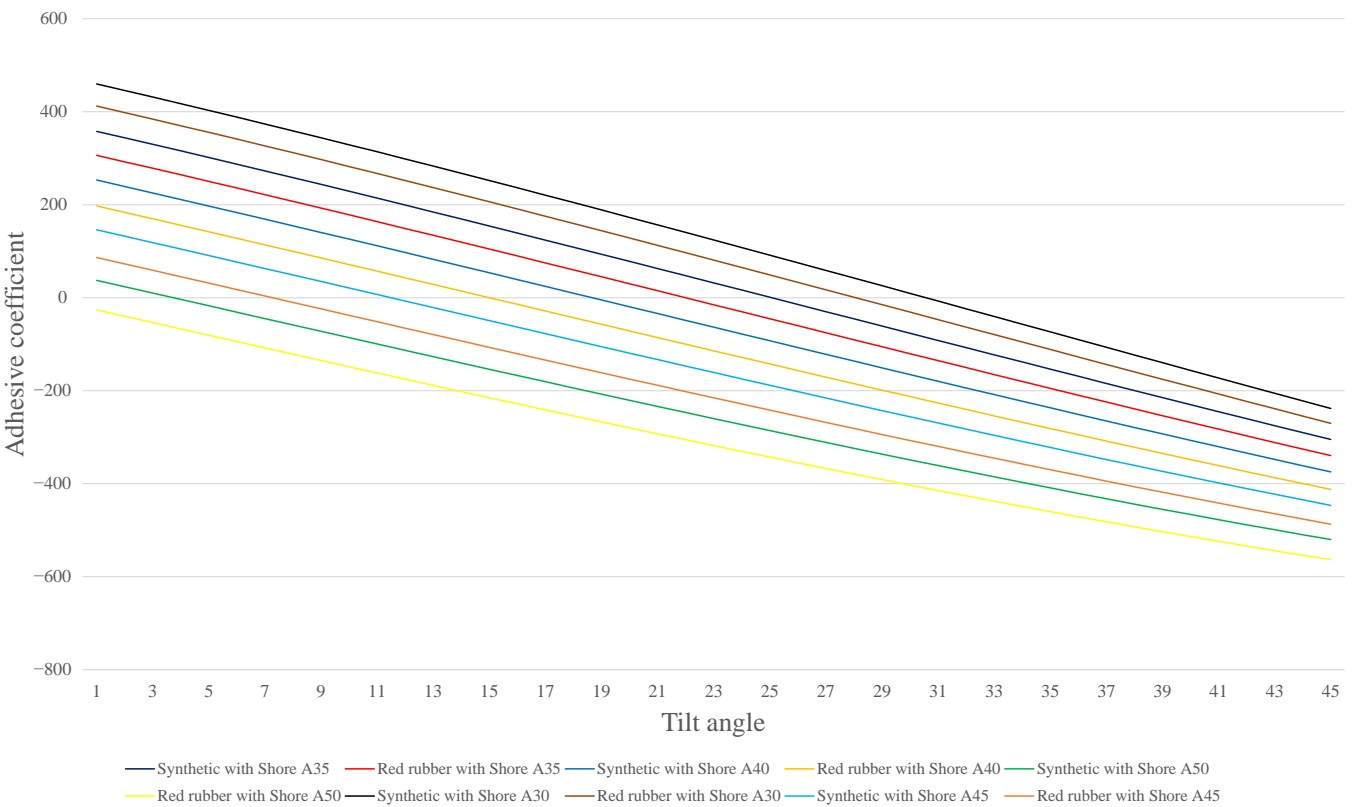

**Figure 7.** Relationship between adhesive coefficient and tilted angle .

**Table 2.** Maximum value of tilted angle for each case of material and shore hardness.

| Material | Shore Hardness | Maximum Tilted Angle |
|---|---|---|
| Synthetic rubber | Shore A30 | 30° |
| Red rubber | | 28° |
| Synthetic rubber | Shore A35 | 25° |
| Red rubber | | 21° |
| Synthetic rubber | Shore A40 | 18° |
| Red rubber | | 15° |
| Synthetic rubber | Shore A45 | 11° |
| Red rubber | | 7° |
| Synthetic rubber | Shore A50 | 3° |
| Red rubber | | 0° |

Figure 8 shows the characteristic of shore hardness and the adhesive coefficient for synthetic/red rubber in the case of the tilted angle increment 5° from 15° ÷ 30°. The maximum shore hardness that enables the cleaning robot to move without slippage corresponding to each increment step of the tilted angle is briefly described in Table 3.

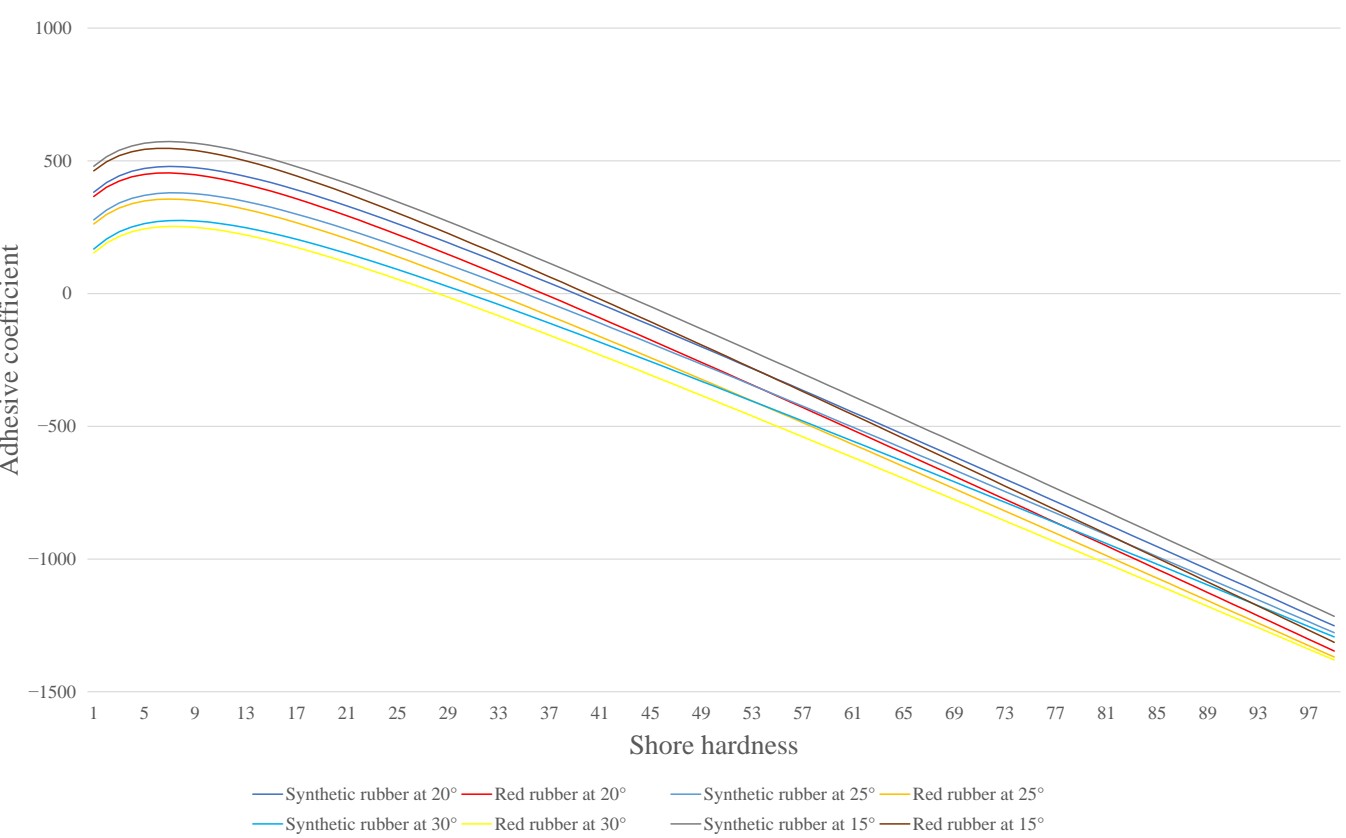

**Figure 8.** Relationship between adhesive coefficient and shore hardness.

**Table 3.** Maximum value of shore hardness for each case of material and tilted angle.

| Material | Tilted Angle | Maximum Shore Hardness |
| --- | --- | --- |
| Synthetic rubber | 30° | A30 |
| Red rubber | | A28 |
| Synthetic rubber | 25° | A35 |
| Red rubber | | A32 |
| Synthetic rubber | 20° | A39 |
| Red rubber | | A36 |
| Synthetic rubber | 15° | A42 |
| Red rubber | | A40 |

It can be seen from Figure 8 and Table 3 that the harder the shore hardness, the lower the adhesive coefficient. It can be realized that the shore hardness reaches over A42 for the synthetic rubber, leading to the robot slippage in the 15° tilted angle case, and the best adhesive of the robot when the belt hardness is Shore A7 for both rubbers. However, the rubber track belt made of rubber with low shore hardness is easily destroyed by high pressure. Additionally, it is noted that the shore hardness is proportional to the adhesive coefficient in the transient stage for the shore hardness less than A7. This can be explained by the destruction of the bond structure of rubber material that has reached the compression limit.

## 4. Conclusions

This paper investigated the adhesive coefficient of the rubber wheel crawler with the coating material of synthetic and red rubber on the wet tilted PV by introducing the theoretical model to validate the slippage. The theoretical model was established by using the physical laws and deriving parameters from experiments. The accuracy of the proposed theoretical model was evaluated through the comparison between simulation

and experimental results. Through the theoretical model, it can be pointed out that the shore hardness of A7 can provide the biggest adhesive coefficient; however, it is hard to manufacture the rubber track belt with a durable lifetime. The shore hardness over A40 can cause the slippage of the rubber wheel crawler. For a specified application, the anti-slippage of the rubber wheel crawler can be verified by the proposed theoretical model. For a different tilted angle of the shopfloor, robot weight, and track belt material, one can seek the optimal value for a certain case. In this paper, some assumptions are given to simplify the consideration of the adhesive coefficient. The limitation of this paper is that the adhesive coefficient is considered under some given assumptions. First, the contact area between the rubber wheel crawler and the PV surfaces is uniform at all points. Then, the drive efficiency of the rubber track belt and the timing pulley is ignored. Next, the belt tension and the contact area of the rubber wheel crawler and the PV surfaces are the same as the two sides of the cleaning robot. Finally, the affection of the rotating brush to the robot slippage is ignored.

This paper mainly focuses on the shore hardness of the coating material on the rubber wheel crawler affecting the adhesive coefficient when the cleaning robot moves on the wet PV surface. For future research, the structure of coating the trackpad with a deep-recessed pattern is an interesting topic.

**Author Contributions:** Data curation, M.T.N. and V.T.N.; Formal analysis, C.T.T.; Project administration, H.H.N.; Supervision, T.T.N.; Writing—original draft, M.T.N. and C.T.T.; Writing—review & editing, V.T.D. and T.T.N. All authors have read and agreed to the published version of the manuscript.

**Funding:** This research is funded by Vietnam National University Ho Chi Minh City (VNU-HCM) under grant number C2021-20-10.

**Institutional Review Board Statement:** Not applicable.

**Informed Consent Statement:** Not applicable.

**Data Availability Statement:** Data sharing is not applicable to this article.

**Acknowledgments:** This research is funded by Vietnam National University Ho Chi Minh City (VNU-HCM) under grant number C2021-20-10. We acknowledge the support of time and facilities from the Nation Key Laboratory of Digital Control and System Engineering (DCSELab), Ho Chi Minh City University of Technology (HCMUT), VNU-HCM for this study.

**Conflicts of Interest:** The authors declare no conflict of interest.

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
