# Peer review of "Research on Adhesive Coefficient of Rubber Wheel Crawler on Wet Tilted Photovoltaic Panel"

_applsci, doi:10.3390/app12136605_

Round 1

Reviewer 1 Report

The article is a very relevant subject and written at a high level.
However, there are some points that need to be corrected, so I recommend a major revision of the article.

The title of the article completely corresponds to it.
However, the article has a few points that need to be corrected:
·        The style of presenting graphical results needs to be changed. (black for numbers and other meanings).
·        The authors make a mathematical model and present an experimental setup. However, no comparisons are made of the results obtained. This should show how much the theoretical model has an error in the actual results. The author must prove that the theoretical results coincide with the real materials.
·        Did the authors take into account the influence of mobile robot motion parameters on the obtained results? How were the experimental results (methodology) conducted? What measuring instruments were used and parameters of materials were determined? All this must be described.
·        There are many design errors in the article (degree notation, number ranges, etc.)
·        The conclusion does not refer to formulas or literature, this should be corrected.

Author Response

We thank you for your time and patience in reviewing our manuscript. Based on your helpful suggestions, we have thoroughly revised our manuscript and provided detailed clarifications in response to the comments.

All the revisions are highlighted in the manuscript in yellow marker. The detailed responses to the editor and reviewer are attached in the “Supporting Document” section below the main document. Except for the above revisions, we also corrected the detailed mistakes, for example, typos and inappropriate wording. The detailed responses to the editor and reviewer are listed below, please kindly find it.

Thank you again for your helpful comments and suggestions that helped us to revise our manuscript. We would be glad to respond to any further questions and comments that you may have.

Best regards,

Authors

Reviewer 2 Report

Introduction : Please add how the nuclear energy may harm humans.?  Nuclear issues are not specific to humans.  If there is a catastrophe it impacts, plant and animal life.  (line 26-27)

The materials and methods needs work.  please review the equations.

Eg: Equation 12 do you mean FSA or FSB ?

Experimental study:  figure 7.  the tilt and the adhesive co-eff seem very linear.  Is this accurate?  

The research itself a bit confusing.  Based on the title and the abstract I was hoping to see a type of material/ some form of rubber that provides suction from outside.  However, I see that the robot in figure 6 navigates based on the locks / grooves in the belt.  

There have been many research done on the track robots.  The implementation is good in this research.  However, I feel that the robot used for cleaning seems to be heavy based on personal experience.  I recommend.

Author Response

(The authors gave the same response as above.)

Round 2

Reviewer 1 Report

The authors of the article corrected many minor mistakes and improved the article. However, questions remain about the adequacy of the results obtained, as they are only theoretical. In view of this, the authors need to change the title, annotation and main points of the article with a focus on theoretical research. Therefore, I recommend a minor revision of the article.

Author Response

(The authors gave the same response as above.)
